# Study on Impoundment Deformation Characteristics and Crack of High Core Rockfill Dam Based on Inversion Parameters

Litan Pan [1], Bo Wu [2,3], Daquan Wang [1], Xiongxiong Zhou [2,3,*], Lijie Wang [1] and Yi Zhang [4]

[1] Huadian Electric Power Research Institute Co., Ltd., Hangzhou 310000, China; daquan-wang@chder.com (D.W.)

[2] College of Water Resources and Architectural Engineering, Northwest A & F University, Xianyang 712100, China

[3] Key Laboratory of Agricultural Soil and Water Engineering in Arid and Semiarid Areas of Ministry of Education, Northwest A & F University, Xianyang 712100, China

[4] Dagu Hydropower Branch of Huadian Xizang Energy Co., Ltd., Shannan 856000, China

\* Correspondence: zhouxx@nwafu.edu.cn

**Abstract:** In the numerical simulation of earth-rock dam, accurate and reliable mechanical parameters of the dam material are the important basis for dam deformation predictions and dam safety evaluations. Based on the deformation monitoring data of Luding core wall rockfill dam, the rheological parameters of rockfill and core wall materials are inverted in this paper. Combined with the actual filling and impoundment process of the dam, the numerical simulation is carried out, and the stress deformation and differential settlement of the dam after completion and impoundment are analyzed. The results showed that the stress deformation results of the dam based on the inversion parameters were in good agreement with the actual deformation. The horizontal displacement, settlement, and principal stress of the dam during the completion period were symmetrically distributed along the core wall. The maximum horizontal displacement occurred at the main dam on both sides of the core wall and the upstream and downstream dam slopes, and the maximum settlement occurred in the middle of the core wall. During the impoundment period, under the action of reservoir water pressure and upstream rockfill wetting deformation, the deformation and stress of the dam body no longer met the symmetrical distribution law, and the maximum horizontal displacement of the dam body during the impoundment period was located at 2/3 of the upstream dam slope. The maximum settlement of the dam body was located at 1/2 of the dam height. The maximum principal stress on the upstream side of the core wall was located on the left side of the bottom of the core wall, and the minimum principal stress was also located on the left side of the bottom of the core wall. The simulation results of the deformation and stress met the general law of earth-rock dam engineering. During the completion period, the deformation inclination of the dam crest was less than 1%. During the impoundment period, the deformation inclination of the dam crest area increased due to the wetting deformation of the upstream rockfill material. At the same time, the deformation inclination of the dam crest axis was larger than that of the upstream and downstream sides, and the deformation inclination of the dam crest at the middle of the valley was the largest, but it did not exceed 3%, that is, there would be no longitudinal cracks, which is consistent with the actual situation. The research results can better predict the stress deformation and crack of the dam body, and provide important support for dam safety evaluations.

**Keywords:** high core wall rockfill dam; parameter inversion; finite element calculation; deformation inclination; cracks

## 1. Introduction

Deformation stability and control are key technical issues in the construction of high rockfill dams [1,2]. There are many examples of large deformation, uneven settlement, leakage, cracks, and other problems caused by improper deformation control in the existing

high rockfill dams in China, and some even directly cause dam failures [3,4]. At present, there are a large number of high earth and rock dams under construction and preparation in the western region of China [5]. With the increase in the number and magnitude of high rockfill dams, it is increasingly important to analyze stress deformation and cracks to ensure the safe and stable operation of the dam. Obtaining correct and reliable dam material parameters is the foundation for calculation and analysis [6]. The stress deformation analysis of earth-rock dams based on inversion parameters is of great significance for predicting the potential adverse deformation and safety risks that may occur in a dam [7,8].

The deformation characteristics of rockfill materials are complex, and indoor tests are influenced by factors such as stress path limitations and size effects, resulting in parameters that often differ significantly from their actual parameters [9,10]. Therefore, obtaining the rheological parameters of dam materials through parameter inversion is the research object of many scholars [11,12]. The basic idea of inversion analysis of geotechnical mechanical parameters was first proposed by Kavanagh, Gioda, Maier, and others in 1971. In 1981, Gioda [13], an Italian professor, carried out the research work of optimization analysis based on the idea of numerical iteration. The Powell method, simplex method, quasi-gradient method, least squares method, and other optimization methods were used to study the displacement back analysis. In 1983, Sakurai [14] combined a finite element method to carry out a numerical calculation of practical engineering, and the parameters of soil and rock materials were obtained using inversion. After that, Simpano et al. [15] applied the genetic algorithm to inversion analysis, and set the precedent by going from a non-intelligent inversion to a modern intelligent inversion algorithm. In the early 1990s, inversion analysis work in China was gradually carried out. Zhou et al. [16] discussed the plastic characteristics of rheological deformation of rockfill materials under multi-stage loading conditions. Inspired by genetic engineering, Zhou et al. [17] proposed an improved genetic algorithm based on gene fragment difference to solve high-dimensional non-linear inversion problems. Wu et al. [18] studied complex multi-model and multi-parameter problems using the decoupling back analysis method. Zhu et al. [19] studied parameter inversion of the Maopingxi asphalt concrete core rockfill dam based on the immune genetic algorithm. Zhao et al. [20] used the particle swarm inversion algorithm combined with Adina to carry out an inversion analysis and research on the rockfill materials of Nuozhadu core wall rockfill dam. Chen [21] proposed a new settlement prediction model combined with parameter inversion, which had high deformation prediction accuracy for high rockfill dams. Li et al. [22] proposed an FEM-Bayesian kriging (FBK) method which can effectively characterize the deformation behavior of earth dams. Liu et al. [23] accurately analyzed the relationship between concrete strength and various variables by integrating the orthogonal test and neural network method.

Based on the traditional finite element calculation of dams and the deformation monitoring data of Luding dam, the rheological parameters of the core wall and the rockfill area are inverted in this paper. Then, a three-dimensional stress deformation simulation calculation is carried out based on the inversion parameters, and the deformation characteristics and crack development of Luding high core wall rockfill dam are analyzed by using the simulation results. It provides a more complete and scientific method for predicting the behavior of rockfill dam and the idea of crack analysis, which provides experience and reference for subsequent projects.

## 2. Project Overview

The Luding Hydropower Station is located in Luding County, Sichuan Province. It is the 12th level power station developed by the Dadu River main flow elevator, with the main task of generating electricity. The normal storage level of the reservoir is 1378.0 m, with a total storage capacity of 219.5 million $m^3$ and a regulated storage capacity of 22 million $m^3$. It has daily regulation performance and an installed capacity of 920 MW [24].

The project is classified as a second class project, with a scale of large (2). The power station hub mainly consists of water retaining structures, flood discharge structures, water

diversion and power generation structures, etc. The water retaining structures are clay core wall dams, with one flood discharge tunnel on the left bank and two flood discharge tunnels on the right bank. The water diversion and power generation structures are arranged outside the flood discharge system on the right bank of the riverbed.

The dam crest elevation of clay core wall dam is 1385.5 m, the maximum dam height is 84.0 m, the dam crest length is 537.0 m, the upstream and downstream dam slopes are 1:2, and the dam crest width is 12.0 m. The dam body is divided into four areas: core wall, dam shell rockfill, filter layer, and transition layer. The dam body adopts clay core wall for anti-seepage. The riverbed section of the dam foundation adopts a 110 m deep anti-seepage wall connected to curtain grouting, and the anti-seepage plan of closed anti-seepage walls is adopted on both sides. The typical cross-section of the dam is shown in Figure 1.

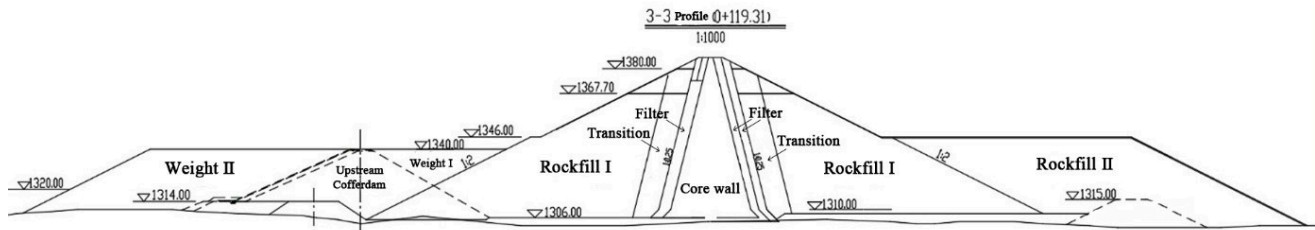

**Figure 1.** Typical profile of Luding core wall rockfill dam.

## 3. Finite Element Simulation

Based on the self-developed static finite element calculation program of earth-rock dams, this paper simulates the filling and impoundment process of the dam. In the numerical simulation, the static deformation, rheological deformation, and wetting deformation under the action of self-weight and water load are mainly considered, which correspond to the constitutive model, rheological model, and wetting model, respectively. In order to better simulate the real deformation form of the core wall dam, this paper attempts to combine the monitoring data and use the intelligent algorithm to invert the mechanical parameters of the dam material. Due to natural factors, the monitoring data of the Luding core wall rockfill dam during the filling and partial impoundment period were lost. We should have inverted the constitutive model parameters to better simulate the true state of the dam. However, due to the lack of data, we had no way to combine the data of the filling and impoundment period to invert the constitutive model parameters. Therefore, we could only combine the existing operating data to invert the rheological parameters, and combine the inverted parameters to approximate the true state of the dam as much as possible.

According to the above research content, the technical route adopted in this paper is shown in Figure 2.

In this paper, the self-developed earth-rock dam calculation program is used to realize the simulation calculation process. The simulation principle flow of the earth-rock dam filling and impoundment process is shown in Figure 3.

### 3.1. Finite Element Model and Simulation Process

The three-dimensional finite element model of Luding clay core rockfill dam was established. As shown in Figure 4, the model mainly uses eight-node hexahedral elements. The total number of model elements was 33,665, and the total number of model nodes was 34,855. The model was divided into 10 material partitions such as core wall, upstream rockfill area, downstream rockfill area, filter layer, and transition layer, as shown in Figure 5.

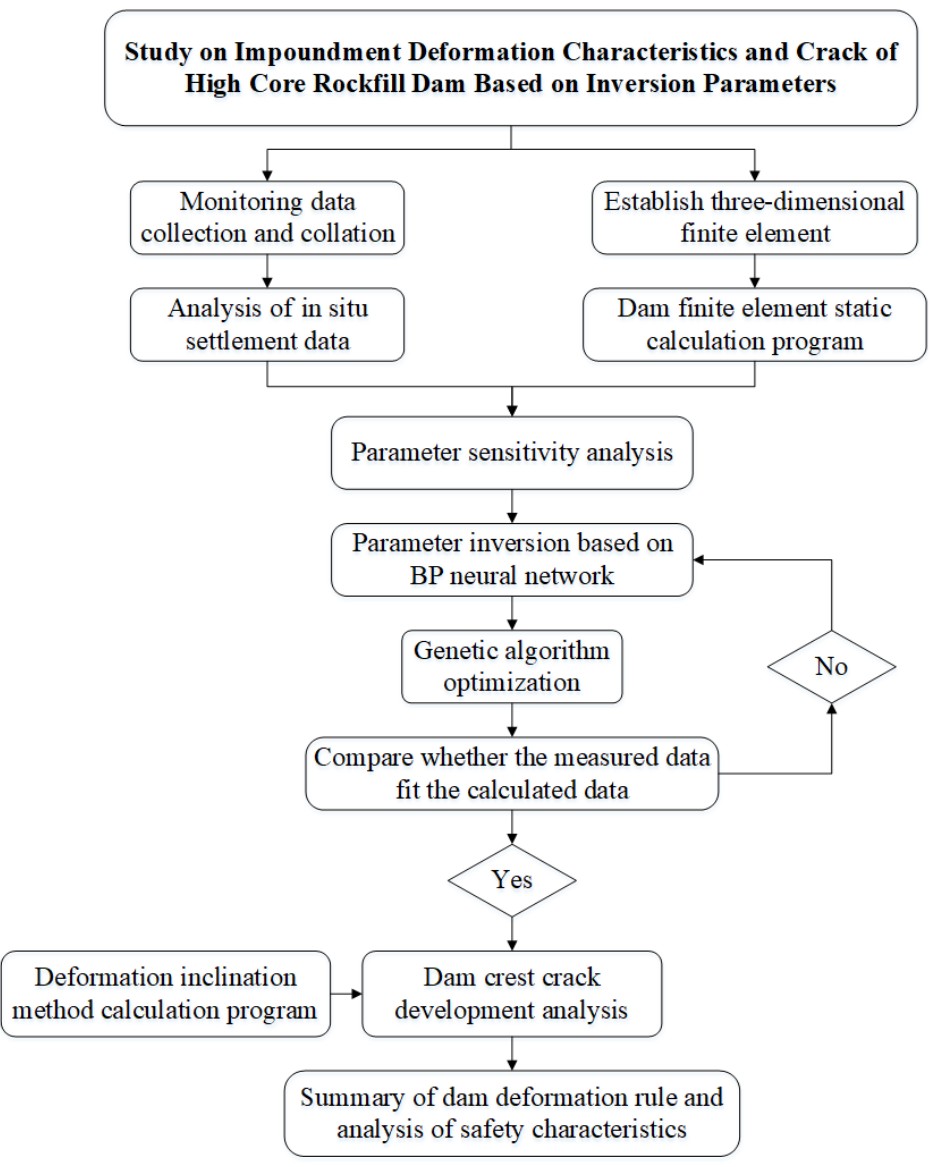

**Figure 2.** Technology roadmap.

This simulation simulates the entire process of dam filling, water storage, and operation. The calculation period was from the date of dam filling to the second half of 2021. The dam construction and water storage simulation were divided into 112 load steps, including 36 load steps during the filling period, 17 water storage load steps, and 69 rheological load steps. On 19 March 2010, the downstream cofferdam was filled to the design elevation; on 31 May 2010, the upstream cofferdam was filled to the design elevation; the dam body filling began in July 2010, and, on 6 March 2011, the lower gate of the 1 # diversion tunnel was sealed; on 25 April 2011, the dam was filled to the design elevation; the water storage began on 20 August 2011, and, after entering the operating period, the long-term deformation of the dam body was simulated and calculated based on the process of reservoir water level changes.

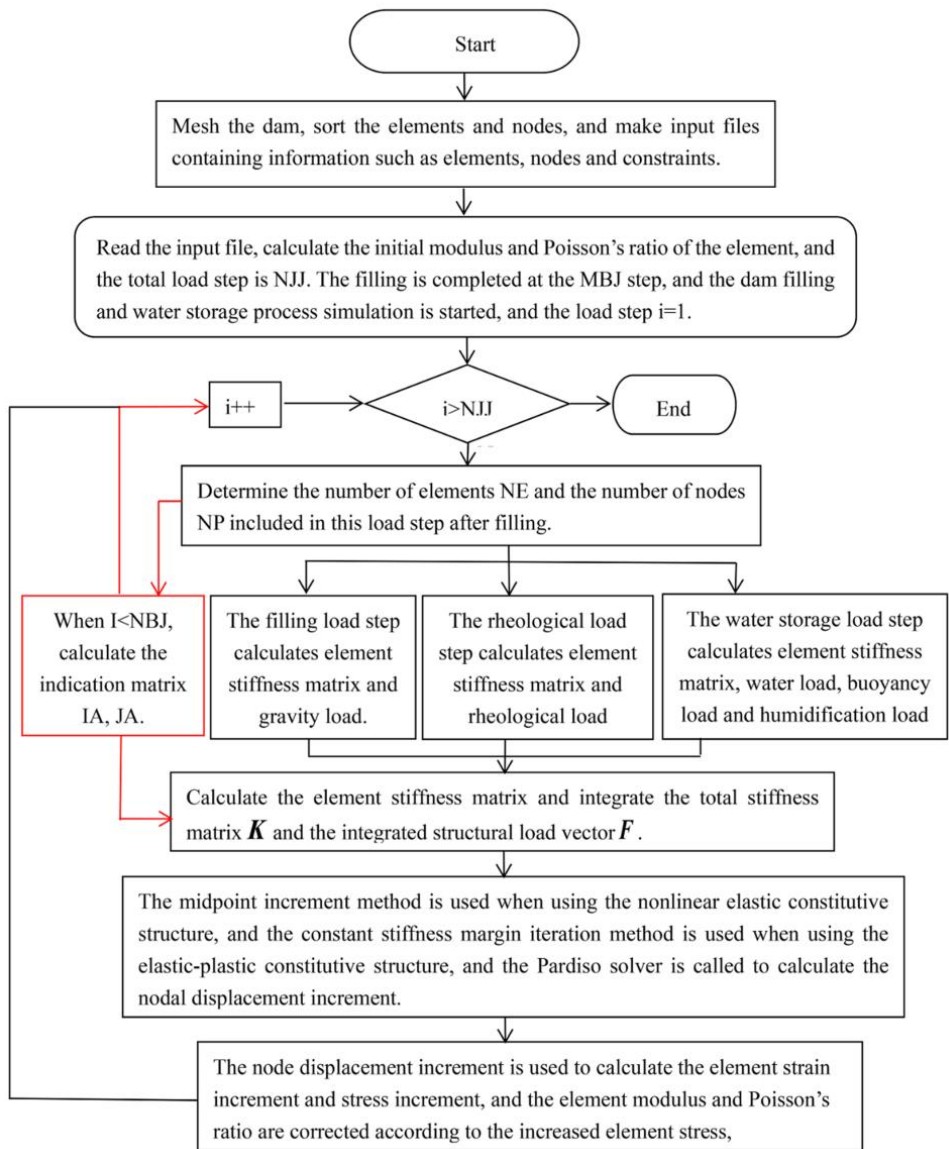

**Figure 3.** A flow chart of the filling process simulation of earth and rockfill dams.

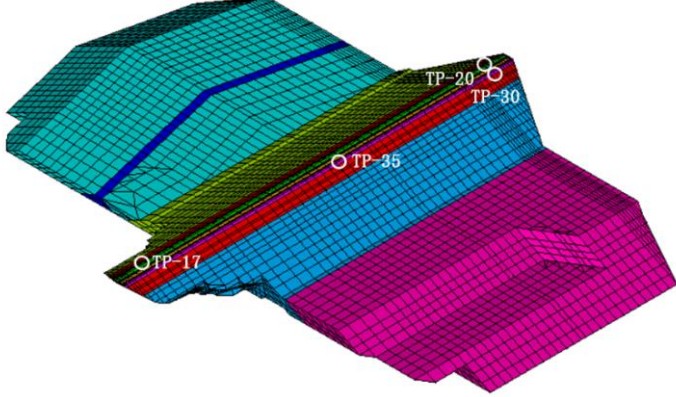

**Figure 4.** Three-dimensional finite element model of Luding dam.

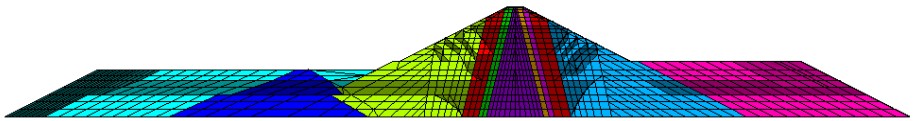

**Figure 5.** Material partition.

### 3.2. Constitutive Model

The Duncan tensor E-v constitutive model [25] was adopted and elastic modulus and tangential Poisson's ratio are expressed as follows:

Initial Elastic Modulus:

$$E_0 = KP_a\left(\frac{\sigma_3}{P_a}\right)^n \tag{1}$$

where $K$ and $n$ are parameters, $P_a$ is atmospheric pressure, and $\sigma_3$ is confining pressure.

Tangent elastic modulus under loading state:

$$E_t = [1 - R_f S]KP_a\left(\frac{\sigma_3}{P_a}\right)^n \tag{2}$$

$$S = \frac{(1 - \sin\varphi)(\sigma_1 - \sigma_3)}{2c\cos\varphi + 2\sigma_3\sin\varphi} \tag{3}$$

where $S$ is the stress level, $R_f$ is the damage ratio, $c$ is the effective cohesion, $\varphi$ is the effective friction angle, and $\sigma_1 - \sigma_3$ is the corresponding deviatoric stress of the test.

Tangential Poisson's ratio:

$$v_t = \frac{G - F\lg\left(\frac{\sigma_3}{P_a}\right)}{(1 - A)^2} \tag{4}$$

wherein

$$A = \frac{D(\sigma_1 - \sigma_3)}{KP_a\left(\frac{\sigma_3}{P_a}\right)^n\left(1 - \frac{R_f(\sigma_1 - \sigma_3)}{(\sigma_1 - \sigma_3)_f}\right)^2} \tag{5}$$

where $G$, $F$, and $D$ are parameters.

The static model parameters of the Luding dam material were obtained through indoor experiments. The parameters of the *E-v* model are shown in Table 1.

**Table 1.** Dam material constitutive model parameters.

| Dam Material | $\gamma$ | $K$ | $n$ | $C$ (10 kPa) | $\Phi$ (°) | $R_f$ | $G$ | $F$ | $D$ |
|---|---|---|---|---|---|---|---|---|---|
| Core wall | 1.68 | 108 | 0.354 | 9.8 | 21.9 | 0.686 | 0.39 | 0.3 | 2.7 |
| Filter | 2.08 | 1309 | 0.476 | 0 | 46.1 | 0.703 | 0.49 | 0.16 | 8.8 |
| Transition | 2.23 | 759 | 0.16 | 0 | 44.6 | 0.71 | 0.22 | −0.04 | 4.7 |
| Rockfill | 2.19 | 1258 | 0.18 | 0 | 46.6 | 0.84 | 0.3 | −0.122 | 5.4 |

### 3.3. Rheological Model

The rheological properties of rockfill materials also have an important influence on the stress and deformation of the dam [26,27]. The commonly used rheological models in China include the Shen Zhujiang three-parameter model [28], Li Guoying seven-parameter model [29], Cheng Zhanlin nine-parameter model [30], and Zhu Sheng seven-parameter model [31]. This paper adopted the Li Guoying seven-parameter model:

$$\varepsilon_t = \varepsilon_f(1 - e^{\alpha t}) \tag{6}$$

$$\varepsilon_{vf} = b\left(\frac{\sigma_3}{P_a}\right)^{m_1} + c\left(\frac{q}{P_a}\right)^{m_2} \tag{7}$$

$$\gamma_f = d\left(\frac{S}{1-S}\right)^{m_3} \tag{8}$$

where $\varepsilon_t$ is the rheological strain at time $t$; $\varepsilon_f$ is permanent flow strain; $e$ is the natural index; $\alpha$ is the rheological rate; $\varepsilon_{vf}$ is the permanent volume flow strain; $\gamma_f$ is permanent shear flow strain; and $\alpha$, $b$, $c$, $d$, $m_1$, $m_2$, and $m_3$ are the seven parameters of the above rheological model, which were obtained using inversion.

### 3.4. Wetting Deformation Model

The wetting deformation has a significant impact on the deformation, seepage, and stability of earth-rock dam engineering. Therefore, this article considered the wetting deformation caused by the initial water storage, and adopted the wetting deformation model and simulation method proposed by Zhou et al. [32].

Wetting axial strain:

$$\Delta\varepsilon_a^w = \frac{\left[K_1\left(\frac{\sigma_3}{Pa}\right) + A_1\right]S_L}{1 - S_L} + K_0\left(\frac{\sigma_3}{P_a}\right)^{m_0} \tag{9}$$

Wetting secant modulus:

$$E^w = \frac{\sigma_1 - 2v^w\sigma_3}{\frac{\left[K_1\left(\frac{\sigma_3}{Pa}\right) + A_1\right]}{1 - S_L} + K_0\left(\frac{\sigma_3}{Pa}\right)^{m_0}} \tag{10}$$

Wetting Poisson's ratio:

$$v^w = c + dS_L \tag{11}$$

where $K_0$, $K_1$, $m_0$, $A_1$, $c$, and $d$ are test parameters.

The parameters of the wetting model of the similar project Guanyinyan dam are shown in Table 2 [32].

**Table 2.** Upstream rockfill wetting model parameters.

| Parameter | $K_0$ | $M_0$ | $K_1$ | $A_1$ | $c$ | $d$ |
|---|---|---|---|---|---|---|
| Upstream rockfill | 0.061 | 0.596 | 0.052 | 0.923 | 0.348 | 0.104 |

### 3.5. Intelligent Inversion of Rheological Parameters

The rheological parameters used in this experiment included seven parameters: $\alpha$, $b$, $c$, $d$, $m_1$, $m_2$, and $m_3$. The upstream and downstream rockfill materials have the same rheological parameters, and the reverse filter layer and filter layer also have the same rheological parameters as the rockfill materials. Therefore, only the rockfill materials and clay core wall materials were subjected to rheological parameter inversion analysis. The initial inversion values were determined based on the experience of selecting rheological parameter values for Guanyinyan rockfill dam materials. Through the orthogonal experimental method for parameter sensitivity analysis, it was found that the parameters with higher sensitivity were $b$, $c$, and $d$, so inversion research was conducted on them. This article used the BP neural network and genetic algorithm for parameter inversion, combined with measured deformation, to obtain rheological parameters for the final simulation calculation [23,33].

In this paper, three sensitive rheological parameters in the rockfill area and three sensitive rheological parameters in the core wall area were combined, and six rheological parameters corresponded to one fitness value. A total of 300 groups of parameter samples were selected for neural network training through orthogonal method and random access method, and the fitting effect of test set, verification set, training set, and prediction model prediction output value was obtained according to the network regression curve, as shown in Figure 6. It can be seen that the mean squared error remained below 0.02. According to the network regression curve, it can be seen that the test set, validation set, training set, and prediction model's predicted output values fit well. By substituting the trained network into the

genetic algorithm for optimization and continuously adjusting and analyzing the data, the following parameters for the genetic algorithm in this article were obtained: the initial population number was 200, the initial elite number was 10, the crossover genetic probability was 0.8, the mutation probability was 0.01, the immigration probability was 0.2, and the termination evolutionary algebra was 177. The rheological parameters of our inversion were obtained using the genetic algorithm optimization, as shown in Table 3:

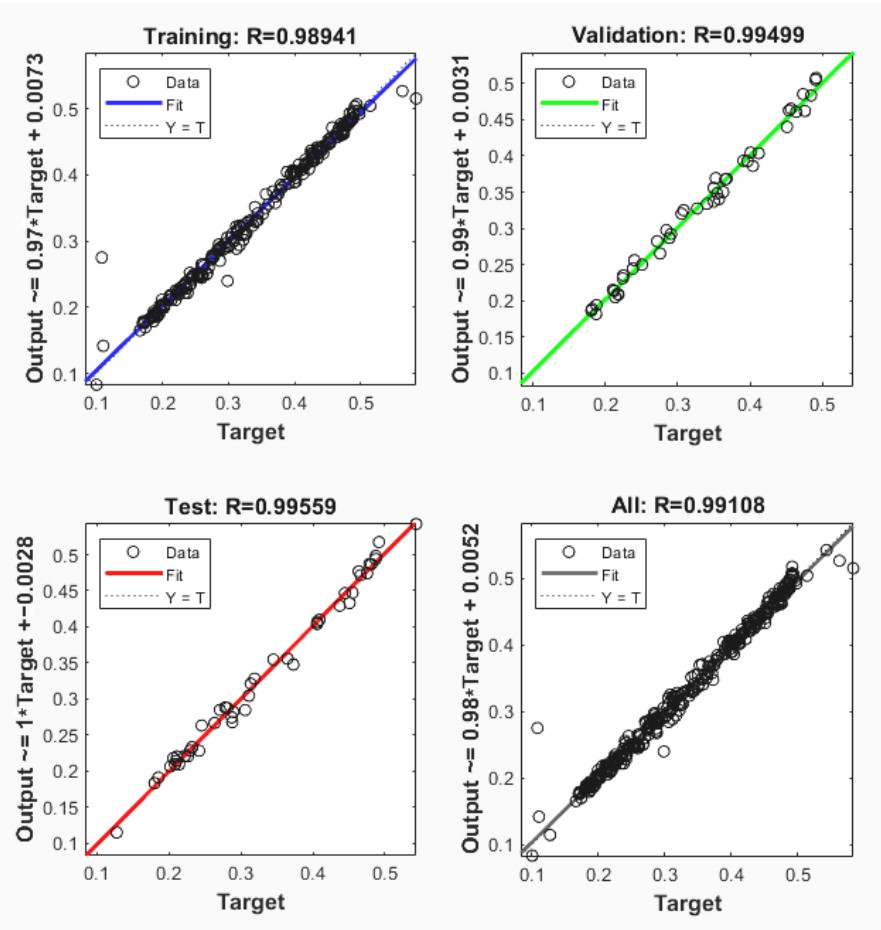

**Figure 6.** Network error regression line.

**Table 3.** Rheological parameters obtained using inversion.

| Model Parameter | $b$ | $c$ | $d$ | $m_1$ | $m_2$ | $m_3$ | $\alpha$ |
|---|---|---|---|---|---|---|---|
| Rockfill area | 0.4 | 0.05 | 1 | 0.301 | 0.3 | 0.4 | 0.0051 |
| Core wall area | 1.426 | 0.2 | 0.225 | 0.6 | 0.9 | 0.302 | 0.0015 |

After the inversion parameters are obtained using the above inversion method, the finite element calculation of Luding earth-rock dam was carried out by using the inversion parameters combined with the four measuring points selected in this paper. The calculated settlement value obtained using the finite element calculation was compared with the measured settlement value of the measured data, as shown in Figure 7. It can be seen that the fitting effect between the calculated value and the measured value was good, so the numerical simulation results were consistent with the actual dam body's behavior, and the simulation results can represent the actual dam body situation.

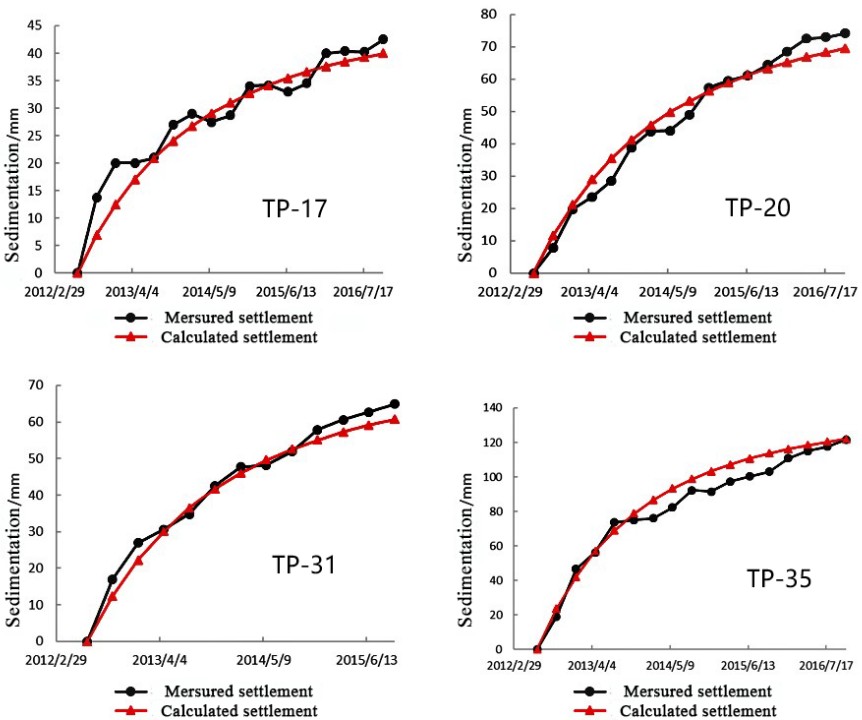

**Figure 7.** Comparison curve between measured settlement and calculated settlement.

## 4. Calculation Results of Stress and Deformation of Dam Body

The stress and deformation of the dam were calculated in the completion period and the impoundment period, and the safety of the dam was evaluated according to the stress and deformation of each position of the dam body. This article selected the 0 + 193 section as the representative research section, and the cloud diagram of the horizontal displacement, settlement, and maximum and minimum principal stresses calculation results under the two working conditions of completion and water storage of this section is shown in Figures 8 and 9. The deformation characteristic values of the dam during the completion period and the impoundment period are shown in Table 4.

**Table 4.** Dam deformation characteristic values.

| Time Limit | Maximum Horizontal Displacement in Upstream Direction/m | Maximum Horizontal Displacement in Downstream Direction/m | Maximum Settlement/m | Maximum Settlement in Proportion to Dam Height/% |
|---|---|---|---|---|
| Completion period | 0.05 | 0.04 | 0.65 | 0.7% |
| Impoundment period | 0.4 | 0.1 | 1.1 | 1.3% |

It can be seen from Figure 8 that the horizontal displacement and settlement of the dam during the completion period were symmetrically distributed along the core wall. The maximum horizontal displacement occurred at the main dam on both sides of the core wall and the upstream and downstream dam slopes, which is about 5 cm. The maximum settlement occurred in the middle of the core wall, which is about 65 cm, or 0.7% of the dam height. The major and minor principal stresses of the dam were also symmetrically distributed along the core wall. The extreme value of the major principal stress occurred on both sides of the bottom of the clay core wall, with a size of about 2.20 MPa, and the extreme value of the minor principal stress occurs on both sides of the bottom of the core wall; this was about 0.67 MPa. The simulation results of the deformation and stress met the general law of earth-rock dam engineering.

Under the action of reservoir water pressure and upstream rockfill wetting deformation, the displacement and stress of the dam body during the impoundment period

changed greatly compared with during the completion period. It can be seen from Figure 7 that the deformation and stress of the dam body no longer satisfy the law of symmetrical distribution. The maximum horizontal displacement of the dam body during the impoundment period was located at 2/3 of the upstream dam slope, which is about 40 cm. The maximum settlement of the dam was located at 1/2 of the dam height, at about 110 cm, accounting for 1.3% of the dam height. The maximum principal stress on the upstream side of the core wall was greater than that on the downstream side, with the extreme value located on the left side of the bottom of the core wall, measuring about 3.00 MPa. The minimum principal stress extreme value was also located on the left side of the bottom of the core wall, measuring about 0.62 MPa. All the simulation results complied with general laws of earth-rock dam engineering.

Comparing Figures 8 and 9, it can be seen that after the dam impoundment, the upstream rockfill material underwent significant changes in both horizontal displacement and settlement due to water storage deformation. The upstream rockfill area experienced significant upstream displacement and settlement, which is highly likely to cause uneven settlement at the dam crest, leading to longitudinal cracks at the dam crest and endangering the safety of the dam. Therefore, it was necessary to calculate and analyze cracks based on the analysis of dam stress and deformation.

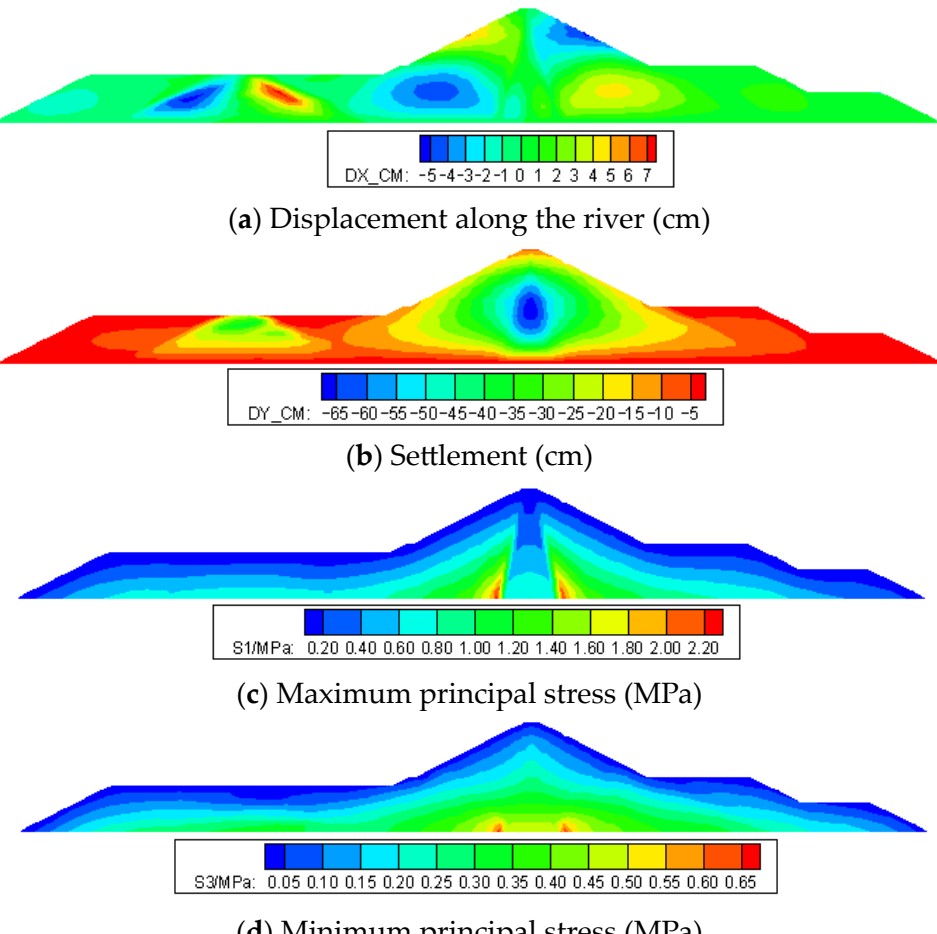

(**a**) Displacement along the river (cm)

(**b**) Settlement (cm)

(**c**) Maximum principal stress (MPa)

(**d**) Minimum principal stress (MPa)

**Figure 8.** Calculation results of completion period.

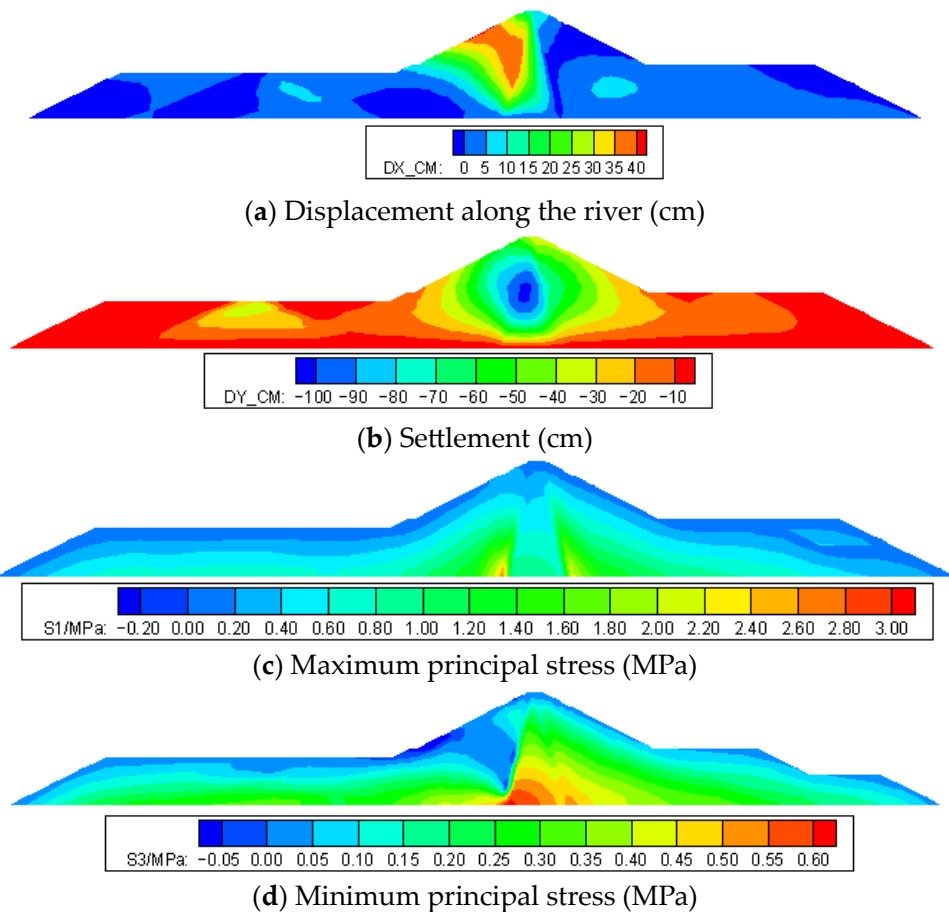

(**a**) Displacement along the river (cm)

(**b**) Settlement (cm)

(**c**) Maximum principal stress (MPa)

(**d**) Minimum principal stress (MPa)

**Figure 9.** Calculation results of impoundment period.

## 5. Fracture Calculation and Analysis

The crack research of earth-rock dams mainly includes two aspects: the crack judgment and the crack propagation simulation [34,35]. The current methods for calculating cracks in earth-rock dams mainly include settlement analysis, the deformation inclination method, Leonard's method, etc. [36–40]. In this paper, the deformation inclination method was used to calculate and analyze the cracks in the Luding core wall earth-rock dam. Figure 10 is the basic principle of the deformation inclination method. $x$ indicates the horizontal direction and $y$ indicates the direction in which the uneven settlement occurs. A and B are two points with a horizontal distance of $\Delta x$. The settlements $S_A$ and $S_B$ occur and move to A′ and B′, respectively. The settlement difference between the two is $\Delta S = S_A - S_B$. The deformation inclination $\gamma$ between the two points is

$$\gamma \approx \tan \gamma = \frac{\Delta S}{\Delta x} \tag{12}$$

Based on experimental and engineering experience, the critical inclination $\gamma_c$ at the time of cracking can be determined. If $\gamma > \gamma_c$, it is considered that cracking occurs, otherwise, it does not occur. For dam building soil and stone materials, the critical inclination is generally taken as 1% or less.

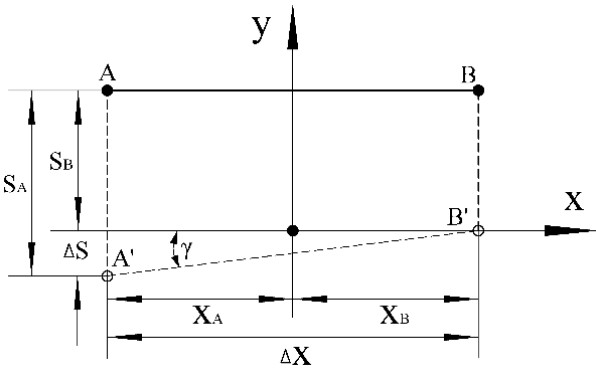

**Figure 10.** Deformation inclination method.

The deformation inclination representation in a finite element simulation is as follows:

$$\begin{cases} \gamma_x = \sum_{i=1}^{k} \frac{\partial N_i}{\partial x} u_{i,y} \\ \gamma_z = \sum_{i=1}^{k} \frac{\partial N_i}{\partial z} u_{i,y} \end{cases} \tag{13}$$

where $k$ is the number of unit nodes, $N_i$ and $u_i$ are the shape function and displacement corresponding to the $i$th node of the element, and $u_{i,y}$ is the displacement of the $i$th node in the $y$ direction.

The Formula (13) for calculating the deformation inclination was written into the finite element calculation program of the earth-rock dam, and the deformation inclination could be calculated directly through the displacement of the dam body after the completion and impoundment of the dam, so as to analyze the cracks.

*Calculation of Dam Deformation Inclination*

Figures 11 and 12 are deformation inclination nephograms of the dam when the filling is completed and after impoundment, respectively. It was found that the overall deformation inclination of the dam body was below 1% when the dam was completed. After impoundment, the deformation inclination near the dam crest was significantly larger than that at the completion of the dam, and the deformation inclination of the dam crest generally exceeded 1%, but did not exceed 3%. According to Gu's research [41], cracks will not occur in earth-rock dams when the deformation inclination is below 1%, and there is a possibility of cracks occurring when the deformation inclination is between 1% and 3%. Cracks must occur when the deformation inclination exceeds 3%. It can be seen that the filling construction quality of the dam meets the standard, and there were no cracks in the dam when the filling was completed; the wetting deformation of the upstream rockfill material after impoundment led to a sharp increase in the deformation inclination of the dam crest, which is a key stage prone to cracks and needs to be focused on.

In order to deeply study the change law of the deformation gradient in the dam crest area, three reference lines parallel to the dam axis running through the left and right banks were selected at the upstream, middle, and downstream sides of the dam crest, respectively, as shown in Figure 13, to study the change law of the deformation gradient on the three lines.

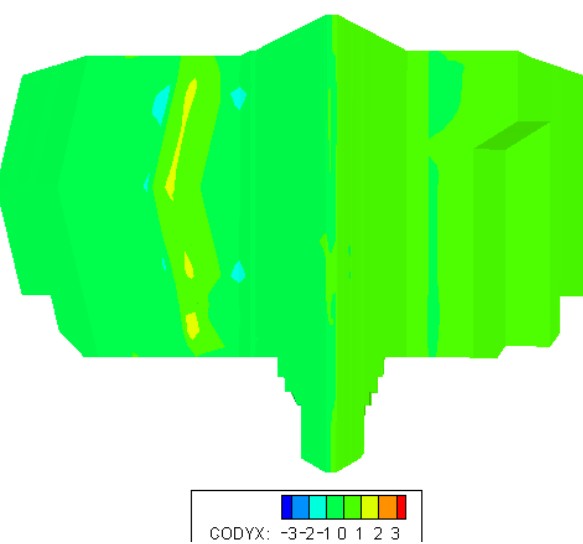

**Figure 11.** Deformation inclination cloud diagram of dam during completion period (%).

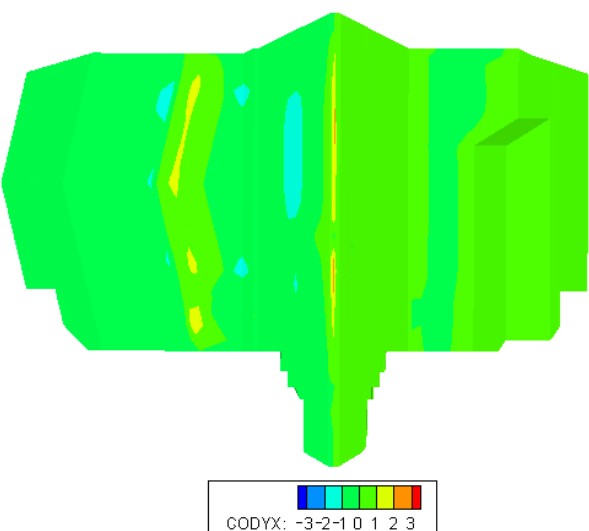

**Figure 12.** Deformation inclination cloud map of dam after impoundment (%).

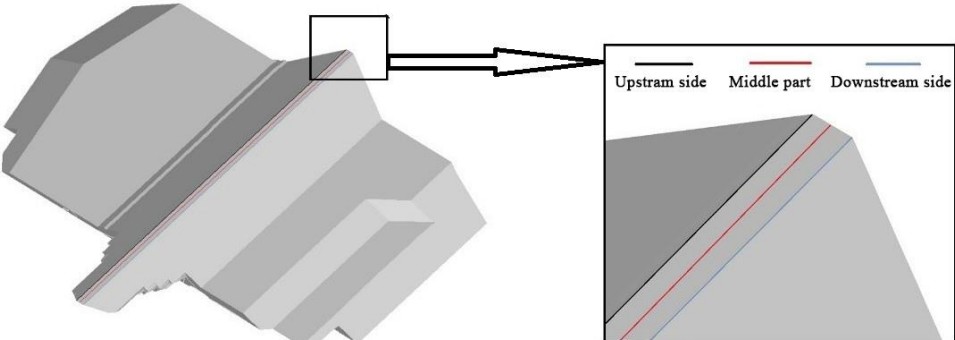

**Figure 13.** Dam top line selection diagram.

The deformation inclination on the three lines during the completion period and the impoundment period is shown in Figures 14 and 15. It can be seen that the deformation inclination of the dam crest during the completion period was below 1%, and there would be no cracks, which is in line with the actual situation. During the water storage period, due to the wetting deformation caused by the flooding of the upstream rockfill area there is

a tendency for the upstream dam slope and shell to detach from the core wall, resulting in an increase in the deformation inclination at the dam crest. However, the value was below 3% and no cracks were generated, which is in line with the actual situation. At the same time, it can be clearly seen that the deformation gradient of the upstream, middle, and downstream sides of the dam crest basically followed the law of large deformation gradients in the middle along the dam axis and small deformation gradients on both banks. In addition, in the simulation results, the deformation inclination in the middle of the dam crest was greater than that on both the upstream and downstream sides of the dam crest, indicating that cracks were more likely to occur in the middle of the dam crest. The reason is that under the action of water storage humidification and water pressure, the upstream rockfill and the core wall are separated, and the dam crest is stretched perpendicular to the dam axis directly above the core wall. This trend is most significant during the initial impoundment period of the dam, and will be alleviated in the subsequent long-term operation. That is, when there are no other factors, the dam does not produce cracks during the initial impoundment period, so there is a high probability that cracks will not occur during the subsequent operation. This shows that the cracks on the dam crest are easily caused by the wetting deformation after the initial impoundment of the earth-rock dam, and the cracks on the dam crest may occur at the dam axis. Analyzing the deformation gradient value of the dam crest has important reference significance for judging whether the dam will produce cracks.

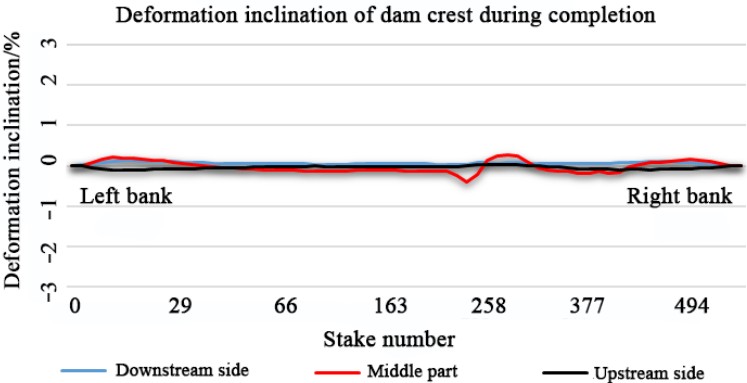

**Figure 14.** Deformation inclination of dam crest during completion (%).

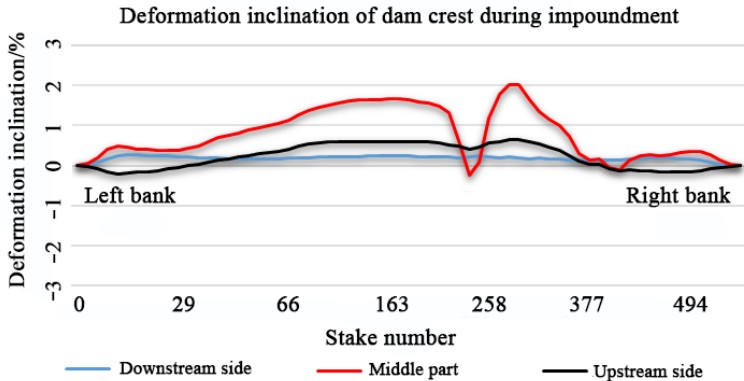

**Figure 15.** Deformation inclination of dam crest during impoundment (%).

## 6. Conclusions and Foresight

### 6.1. Conclusions

This article adopted the core wall rockfill dam of Luding Hydropower Station as the research object, taking into account the rheological and wetting properties of the rockfill material. The finite element method was used to numerically simulate the filling and storage processes of the dam, and the rheological parameters of the key dam materials

were inverted. The actual parameters of the inversion were used to analyze the current stress, deformation, and cracks of the dam through the finite element method, and the stress during the completion and storage periods of the dam were summarized. Based on the characteristics of deformation and cracks, the following conclusions can be drawn:

1. During the completion period, the horizontal displacement, settlement, and major and minor principal stresses of the dam body were symmetrically distributed along the core wall. The maximum horizontal displacement occurred at the main dam body on both sides of the core wall and the upstream and downstream dam slopes. The maximum settlement occurred in the middle of the core wall, the maximum principal stress occurred on both sides of the bottom of the clay core wall, and the minimum principal stress occurred on both sides of the bottom of the core wall. The deformation and stress simulation results met the general laws of earth-rock dam engineering.

2. During the storage period, under the influence of reservoir water pressure and the wetting deformation of upstream rockfill materials, the displacement and stress of the dam body underwent significant changes compared to the completion period. The deformation and stress of the dam no longer followed the symmetrical distribution pattern, and the maximum horizontal displacement of the dam during the water storage period was located at 2/3 of the upstream dam slope. The maximum settlement of the dam body was located at 1/2 of the dam height, the large principal stress on the upstream side of the core wall was greater than that on the downstream side, and the extreme value was located on the left side of the bottom of the core wall, while the extreme value of the small principal stress was also located on the left side of the bottom of the core wall. All the simulation results complied with general laws of earth-rock dam engineering.

3. The deformation inclination of the dam during the completion period was less than 1%, and there would be no cracks formed. After impoundment, due to the wetting deformation of the upstream rockfill area, the deformation gradient of the dam crest changed greatly. After the initial impoundment, the deformation gradient values of the upstream, middle, and downstream sides of the dam crest all increased and were greatest in the middle along the dam axis and perpendicular to the dam axis, that is, the center of the dam crest was the area with the largest deformation gradient, but the maximum value did not exceed 3%. There was a trend of longitudinal cracks along the dam axis, which is consistent with the actual situation. Regular monitoring of cracks in this area is needed to prevent cracks from endangering dam safety.

*6.2. Foresight*

1. This study was limited due to the lack of monitoring data during the dam filling period, so only the rheological parameters were inverted. In subsequent similar dam type research, the dam material Duncan-Chang constitutive model parameter inversion can be carried out in combination with the monitoring data during the filling period. Based on this, the simulation results will be more realistic.

2. In this paper, when using the deformation inclination method to predict cracks, only the linear area of cracks on the surface of the dam top was judged and analyzed. In the follow-up research plan, image recognition technology will be used to analyze the cracks in the whole section of the dam top, which will enable us to analyze the cracks more intuitively and accurately.

**Author Contributions:** Conceptualization, X.Z.; methodology, L.P., B.W. and D.W.; formal analysis, L.P., B.W. and D.W.; resources, L.P., B.W., D.W., L.W. and Y.Z.; writing—original draft preparation, L.P. and B.W.; writing—review and editing, B.W., D.W., X.Z., L.W. and Y.Z.; project administration, X.Z. All authors have read and agreed to the published version of the manuscript.

**Funding:** This research was funded by [the National Natural Science Foundation of China] grant number [52209168] and [the Fundamental Research Funds for the Central Universities of China] grant number [2452020207].

**Data Availability Statement:** Data are contained within the article.

**Acknowledgments:** We are grateful to the target program "Research service of Luding Dam operation state prediction based on full structure online simulation and intelligent analysis of monitoring data" and "Technical services for simulation calculation and parameter inversion of 3 earth-rock dams", which allowed us to complete this work.

**Conflicts of Interest:** Authors Litan Pan, Daquan Wang and Lijie Wang were employed by the company Huadian Electric Power Research Institute Co., Ltd. Author Yi Zhang was employed by the company Dagu Hydropower Branch of Huadian Xizang Energy Co., Ltd. The remaining authors declare that the research was conducted in the absence of any commercial or financial relationships that could be construed as a potential conflict of interest.

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
