# Peer review of "Study on Impoundment Deformation Characteristics and Crack of High Core Rockfill Dam Based on Inversion Parameters"

_water, doi:10.3390/w16010188_

Round 1
Reviewer 1 Report
Comments and Suggestions for Authors
The work is relevant to the journal and can be accepted.
I suggest.
1. Clearly draw the image of the work in the abstract section discussed in the article.
2. A recent literature to the intro. part.
3. A rational paragraph may be added at the end of the introduction part that separates the previous and the present analysis clearly.
4. Cite all equations, where needed.
5. In the results and discussion section, improve the graphical discussion by adding more physical aspects of the results on the state variable. Do not discuss only the trends of the curves.
6. Captions of all the figures need to be updated in the results and discussion section.
7. The conclusion may be adjusted as per the results obtained. Add some comprehensive analysis of the obtained results.
8. The paper may be checked for errors and typos.
Comments on the Quality of English Language
Ok
Author Response
Thank you very much for your suggestions and questions. Answering your questions and implementing your suggestions has given us a deeper understanding of the research of deformation calculation and crack analysis of core wall rockfill dam. According to your suggestions, we have made a lot of modifications to the article, which has greatly improved the readability and academic value of the article. We sincerely thank you for your careful review and pertinent suggestions.

Reviewer 2 Report
Comments and Suggestions for Authors
The paper describes the procedure for selection of material parameters for analysis of long-term performance of Lunding 84-m high rockfill dam. The subject is quiet of interest because cracking the clay core of rockfill dam is a hot topic among dam designers. The reviewer identifies several deficiencies in the current manuscript; therefore, the reviewer suggests that the paper be resubmitted after improvement.
1- The material model formulation, as explained in the paper, is not a true three-dimensional material model. The defined material parameters are selectively set for special case of triaxial tests and major principal strain direction in triaxial test. In the real 3D model, a three-dimensional state of stresses exists, as such extension of the defined stresses from the triaxial test stresses to the general case is not clear.
2- The optimization algorithm for finalizing the material model parameters uses a total of six material parameters (three for rockfill and three for core material) and uses four measured crest deformation with time. Neither the model results of displacement with time nor the measured data supports meaningful change of displacement associated with the wetting deformation. The manuscript does not describe how the wetting parameters are determined and how important are in the matching algorithm.
3- By using six parameters, matching the finite element displacement results with limited measured deformation is rather forgiving and does not necessarily justify neither the validity of calibration nor the model performance. Because there is no inclinometer measured data, it is not feasible to verify the model results for variation of displacement within the body of the dam.
According to the above, the reviewer recommends improving the subscript by:
1- Clarifying the material model for 3D state of stress and how the model implementation and material model satisfies the material model objectivity.
2- Use the measured data before the reservoir impoundment and consider how the dam real pattern of deformation instead of selected discrete crest location can be included in the method.
Author Response
Thank you very much for your suggestions and questions. Answering your questions and implementing your suggestions, has given us a deeper understanding of the research of deformation calculation and crack analysis of core wall rockfill dam. According to your suggestions, we have made a lot of modifications to the article, which has greatly improved the readability and academic value of the article. We sincerely thank you for your careful review and pertinent suggestions.

Reviewer 3 Report
Comments and Suggestions for Authors
The paper focusses on impoundment deformations characteristics and crack of a high core rockfill dam on the basis of inversion parameters. It is a very important issue in the construction of high rockfill dams to ensure safe and stable operation. The object subjected to the research was Luding High Core Rockfill which is located in Sichuan Province in China. In the analysis, the authors take into account the rheological and wetting properties of the rockfill material. Based on the data of deformation monitoring, the rheological parameters of the rock fill and the core wall materials were inverted using genetic algorithm optimisation. After the inversion parameters were obtained, the finite element calculation of the Luding rock dam was carried out using the inversion parameters combined with the four selected measuring points. Finally, the calculated settlement was compared with the measured settlement.
It is very important that the authors proved that the numerical simulation results were consistent with the actual behaviour of the dam structure. The stress and deformation were then analysed in the completion and impoundment period and the safety of the dam was evaluated according to the stress and deformation of each position of the dam structure. Crack research includes crack judgement and crack propagation simulation.
On assessing the paper positively, I have a few comments:
- In order to improve the readability of the paper, a separate section on Methodology should be created. It would be good to create a flow chart.
- There is no information on what FEA software the numerical analysis was performed with.
- The authors’ conclusions are consistent with the evidence and arguments presented in the paper, but should present limitations in addition to the main achievements. Furthermore, there are no proposals for future research.
- The references could be expanded to include more items from outside China.
In conclusion, it should be said that the research carried out is very important in terms of the prediction of stress deformation and cracking of the dam structure and, consequently, for the evaluation of dam safety. The research carried out significantly extends the scientific literature on the problem under consideration. Therefore, I recommend this article for publication in the journal water.
Comments on the Quality of English LanguageModerate editing of English language required.
Author Response
Thank you very much for your suggestions and questions. Answering your questions and implementing your suggestions, has given us a deeper understanding of the research of Stress deformation calculation and crack analysis of core wall rockfill dam. According to your suggestions, we have made a lot of modifications to the article, which has greatly improved the readability and academic value of the article. We sincerely thank you for your careful review and pertinent suggestions.

Round 2
Reviewer 2 Report
Comments and Suggestions for Authors
The revised manuscript does not address my earlier review comments. The description of material model is vague. The type and number of measured data for verification are not sufficient to capture the suggested model performance. This is a very interesting topic, however the manuscript without addressing the comments does not provide a better understanding of the subject.
Comments on the Quality of English LanguageEnglish language is fine.
Author Response
Thank you very much for your suggestions and questions. Answering your questions and implementing your suggestions, have given us a deeper understanding of the research of Stress deformation calculation and crack analysis of core wall rockfill dam. According to your suggestions, we have made a lot of modifications and considerations to the manuscript, which has greatly improved the readability and academic value of the article. We sincerely thank you for your careful review and pertinent suggestions. In the follow-up study, we will continue to carry out in-depth research, and have added future prospects in the text based on your opinions. We apologize that we were unable to fully implement the modifications based on your comments and inquiries, but we have made every effort and hope for your understanding.

Round 3
Reviewer 2 Report
Comments and Suggestions for Authors
The material model can be improved. It appears it is not a function of intermediate principal stress in oppose to soil material that should be pressure dependent.